# Energy landscapes of peptide-MHC binding

**Laura Collesano**[1], **Marta Łuksza**[2], **Michael Lässig**[1]*

**1** Institute for Biological Physics, University of Cologne, Cologne, Germany, **2** Tisch Cancer Institute, Departments of Oncological Sciences and Genetics and Genomic Sciences, Icahn School of Medicine at Mount Sinai, New York, New York, United States of America

* mlaessig@uni-koeln.de

## Abstract

Molecules of the Major Histocompatibility Complex (MHC) present short protein fragments on the cell surface, an important step in T cell immune recognition. MHC-I molecules process peptides from intracellular proteins; MHC-II molecules act in antigen-presenting cells and present peptides derived from extracellular proteins. Here we show that the sequence-dependent energy landscapes of MHC-peptide binding encode class-specific nonlinearities (epistasis). MHC-I has a smooth landscape with global epistasis; the binding energy is a simple deformation of an underlying linear trait. This form of epistasis enhances the discrimination between strong-binding peptides. In contrast, MHC-II has a rugged landscape with idiosyncratic epistasis: binding depends on detailed amino acid combinations at multiple positions of the peptide sequence. The form of epistasis affects the learning of energy landscapes from training data. For MHC-I, a low-complexity problem, we derive a simple matrix model of binding energies that outperforms current models trained by machine learning. For MHC-II, higher complexity prevents learning by simple regression methods. Epistasis also affects the energy and fitness effects of mutations in antigen-derived peptides (epitopes). In MHC-I, large-effect mutations occur predominantly in anchor positions of strong-binding epitopes. In MHC-II, large effects depend on the background epitope sequence but are broadly distributed over the epitope, generating a bigger target for escape mutations due to loss of presentation. Together, our analysis shows how an energy landscape of protein-protein binding constrains the target of escape mutations from T cell immunity, linking the complexity of the molecular interactions to the dynamics of adaptive immune response.

## Author summary

T cell immunity involves the binding of short peptides to the intracellular MHC recognition machinery. Understanding how the peptide-MHC binding energy depends on the peptide sequence is key to computationally predict immune recognition and immune escape evolution, for example, of pathogens and cancer cells. By analyzing experimental binding affinity data, we find that peptide-MHC interactions are described by nonlinear energy landscapes that depend on the recognition pathway: smooth and easy to learn for MHC class I, rugged and difficult to learn for MHC class II. The different forms of nonlinearity—or epistasis—suggest the most suitable strategy to infer these landscapes. In

**Data Availability Statement:** All curated data, raw data sources, and materials used in this study are available in Supporting information, at: https://github.com/lcollesano/MHC-gem and in the IEDB Analysis Resources at http://tools.iedb.org/mhci/download.

**Funding:** This work was supported by Deutsche Forschungsgemeinschaft SFB 1310 (to M.Lässig). M.Łuksza was supported in part as a Pew Biomedical Scholar and by the Pershing Square Sohn Prize. The funders have not played any role in the study design, data collection and analysis, decision to publish, or preparation of the manuscript.

**Competing interests:** The authors have declared that no competing interests exist.

addition, the nonlinearity has direct implications on the energy and fitness of single mutations, generating distinct distributions of possible targets for immune escape. Together, this work establishes links between biophysical origin, nonlinear structure, learnability from data, and biological implications of peptide-MHC interactions, which can be applicable also to other protein interaction landscapes.

## Introduction

The antigen presentation pathway is pivotal for immune surveillance [1, 2]. Class I and class II Major Histocompatibility Complex (MHC) enable this process by binding and presenting small peptides to CD8+ and CD4+ T cells, respectively. Although there are several steps between the processing of an antigen and its presentation [3], peptide-MHC complex formation is considered the most discriminating stage [4]. Therefore, a peptide with high binding affinity has also a higher chance of being presented and recognized by the immune system [5]. Quantitatively understanding protein-MHC binding is therefore a critical step for vaccine design and cancer immunology [6–8].

In the context of cytotoxic T cell recognition, the antigen is presented on the cell surface in a complex with a MHC-I molecule. The peptides able to bind MHC-I are short chains, on average 8 to 11 amino acids long, resulting from the degradation of intracellular proteins carried out by the proteasome [9, 10]. MHC-II acts in specific antigen-presenting cells, presenting peptides transported from the extracellular space for interaction with helper T cells [11]. The amino acid chains that bind to MHC-II are more variable in length, on average 15–20 amino acids, and the relevant binding occurs in a subset of about 9 amino acids [12, 13]. MHC-I and MHC-II have different binding mechanisms. The MHC-I binding pocket is closed at both ends, causing peptides to be anchored at their extremities, usually in positions 2 and 9, while a weaker interaction acts in the center [14, 15]. In contrast, MHC-II has an open binding pocket, allowing the peptide to be anchored at multiple positions [12, 16].

The complexity of interactions between binding positions represents the key challenge for affinity models [17, 18]. Initial studies on MHC-I include pairwise interactions among specific peptide positions in otherwise additive models [17, 19].

In contrast, state-of-the-art computational tools based on neural networks trained on experimental data contain pervasive nonlinearities in the binding energy function, albeit implicit in the model output. These models are highly effective for MHC-I alleles [20–27], they are also increasingly successful in addressing the higher complexity of the binding mechanism for MHC-II alleles [13, 18, 28–30].

However, given their high number of parameters and their complexity, training becomes difficult for rare alleles insufficiently represented in empirical datasets. Despite their performance in binding affinity prediction, the usage of neural networks can limit the understanding of the relationship between binding energy and peptide sequence, because we lack a mechanistic interpretation of the model output [31].

This paper presents a systematic study of the energy landscapes of protein-MHC binding, which map a given peptide sequence onto the binding affinity to a given MHC allele. These landscapes have nonlinearities, or epistasis [32–37], caused by interactions of multiple peptide and MHC amino acids—a common feature of peptide-MHC binding [17]. The form of epistasis, however, depends on the MHC class. For MHC-II, we find rugged landscapes where the binding energy depends on specific amino acid combinations of a peptide in a network of several binding positions. The MHC-I landscapes are much simpler: the binding energy is a

nonlinear function of an intermediate trait that is additive in the peptide sequence [34], a less complex alternative to the pairwise interactions approach, already well studied in previous works [19]. In the literature on evolution, these two types of epistasis, characterizing MHC-II and MHC-I energy landscapes, are referred to as idiosyncratic and global, respectively [38–41].

In the first part of the paper, we infer global and idiosyncratic epistasis in energy landscapes from empirical data of several MHC-I and MHC-II alleles. We show that data and predictive models for MHC-I and MHC-II have low and high complexity, respectively. In the second part, we discuss two important consequences of the form of epistasis. First, we infer a matrix energy model for MHC-I that outperforms state-of-the-art machine-learning models (NetMHC 4.0), providing predictions of very similar quality with a drastically reduced number of model parameters. This model can be learned by simple regression, demonstrating that the complexity of epistasis sets the optimal learning algorithm. Second, we infer spectra of mutational effects in energy landscapes for MHC-I and MHC-II, and we discuss the differences between these spectra induced by the form of epistasis.

## Results

### Datasets of peptide-MHC binding affinities

For training and testing, we analyze peptide sequences and MHC binding data from the Immune Epitope Database (IEDB) [42]. The datasets are curated before training (Material and methods). For MHC-I, we include the most frequent alleles of HLA-A and HLA-B: HLA-A*02:01, HLA-A*11:01, and HLA-B*07:02 with 6953, 1981, and 1024 curated peptides of length 9 aa. The resulting dataset is similar but not identical to the data used for benchmarking and retraining of NetMHC 4.0 [21, 43]. For MHC-II, our dataset contains 7434 peptides from allele HLA-DRB1*01:01 of length 15 aa with a binding core of length 9 aa. In these data, binding affinities are reported as half-inhibitory concentration (IC50), other peptides are annotated qualitatively as binding or non-binding. Throughout this paper, we describe binding affinities by the free energy $\Delta G = \log(IC50/\rho_0)$, which is measured in units of $k_B T$ and has its zero point at the commonly used binding threshold for strong binders, $\rho_0 = 500$nM (MHC-I) and $\rho_0 = 1000$nM (MHC-II) [44, 45].

### Inference of a linear energy model

Fig 1A shows position-weight matrices, or binding motifs, for two representative alleles HLA-A*02:01 for MHC-I and HLA-DRB1*01:01 for MHC-II. These matrices record the frequencies $p_i(a)$ of amino acid $a$ at a given position in the binding core ($i = 1, \ldots, \ell$), evaluated in the set of binding peptides ($\Delta G \leq 0$). The resulting position-specific sequence information, $\Delta S_i = \sum_a p_i(a) \log(p_i(a)/p_0(a))$, is defined as the Kullback-Leibler distance between the observed amino acid frequency distribution and a null distribution (here $p_0(a) = 1/20$). For MHC-I, the sequence information is concentrated in positions 2 and 9, the known anchor positions of peptide-MHC binding [46] (Fig 1B). In contrast, MHC-II has a more homogeneously distributed sequence information, i.e., less bias towards the anchor positions 1, 4, 6, and 9. The total additive information, $\Delta S = \sum_{i=1}^{\ell} \Delta S_i$, which measures the power of linear sequence models to discriminate peptides presented on the cell surface. We find $\Delta S = 4.5$ for MHC-I and a substantially lower value, $\Delta S = 2.4$, for MHC-II. This suggests that an additive model is insufficient for MHC-II; that is, the discrimination of presented peptides depends on interactions (epistasis).

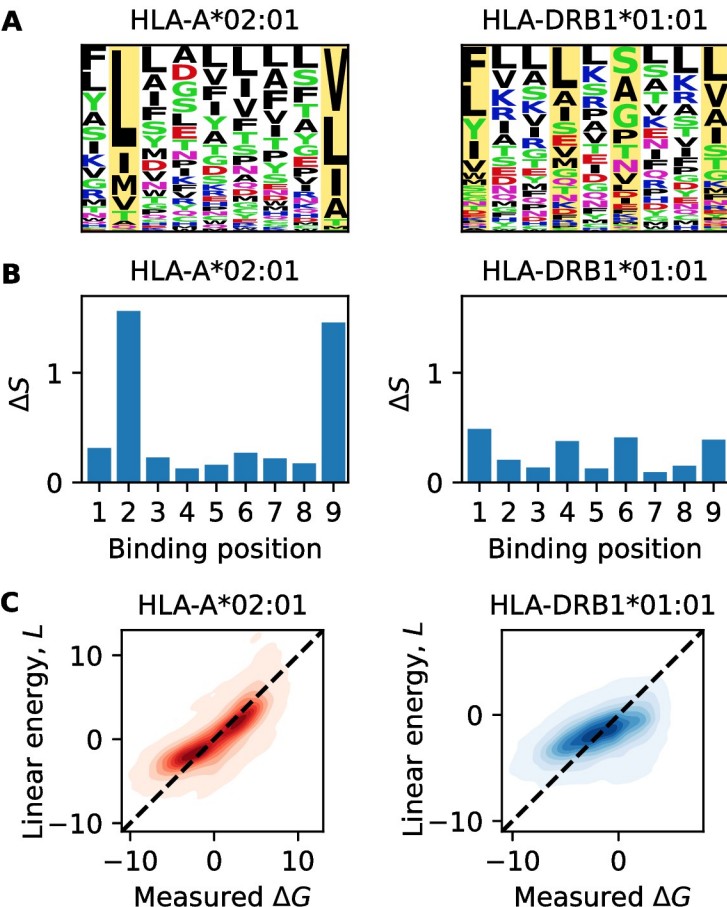

**Fig 1. Binding motifs, sequence information, and linear model.** A: Position weight matrix for HLA-A*02:01 (left) and HLA-DRB1*01:01 (right) inferred from training data. Letter size indicates the frequency of a given amino acid at a given position, $p_i(a)$, in the sample of MHC-binding peptides ($\Delta G \leq 0$). Yellow bars highlight anchor positions, and the color of letters shows the chemical properties of the amino acids (black: hydrophobic, red: acidic, blue: basic, green: polar, purple: neutral) [51]. B: Position-specific sequence information, $\Delta S_i$, defined as the Kullback-Leibler distance of the observed distribution $p_i(a)$ from an equidistribution. C: Error plot comparing observed energies, $\Delta G$, and predictions of the linear model, $L$ (contours give densities above 0.01).

As a starting point for the inference of energy models, we construct a linear energy matrix,

$$L(\mathbf{a}) = \sum_{i=1}^{\ell} \epsilon_i(a_i), \qquad (1)$$

where $\mathbf{a} = (a_1, \ldots, a_\ell)$ denotes the peptide sequence and $\epsilon_i(a)$ is the energy contribution of amino acid $a$ at position $i$. Energy models of this kind are widely used for transcription factor binding and for protein-protein binding, including early-stage predictions of peptide-MHC-I interactions [19, 47]. Here we obtain an optimized $20 \times 9$ energy matrix, which minimizes the mean squared error (MSE), via 10-fold cross-validation for each of the selected alleles (Material and methods). At each position, the energy ranking of amino acids reproduces the frequency ranking in the position weight matrix (S2 and S8 Tables).

The heat maps of Fig 1C show the optimized linear energy $L$ plotted against the validation data for the two representative alleles. For MHC-I, the linear model systematically

underestimates the affinity of strong binders ($L > \Delta G$). A similar effect is observed for the weakest binders, albeit supported by fewer data points. For MHC-II, the linear model has a more drastic mismatch with the data and only explains about half of the variation in $\Delta G$. Likely factors contributing to this mismatch include the variation of binding cores and additional energy contributions of peptide flanking regions not included in the model [48–50]. In what follows, we will complement the linear model by epistasis, the form of which depends on the MHC class.

## Energy landscapes with global and idiosyncratic epistasis

Nonlinear energy landscapes provide a better representation of peptide-MHC binding. Here we refer to such nonlinearities as energy epistasis and parametrize them by interaction terms involving pairs of amino acids in the peptide binding core. For MHC-II, we use a generic landscape of the form

$$\Delta G(\mathbf{a}) = \sum_i \epsilon_i(a_i) + \sum_{i>j} \beta_{ij}(a_i, a_j); \qquad \text{(MHC-II)} \qquad (2)$$

for each pair of positions, the $20 \times 20$ matrix $\beta_{ij}(a, b)$ characterizes the interaction energy between amino acid $a$ at position $i$ and $b$ at position $j$. For MHC-I, we use a simpler landscape,

$$\Delta G(\mathbf{a}) = \sum_i \epsilon_i(a_i) + \lambda \sum_{i,j} \epsilon_i(a_i)\epsilon_j(a_j); \qquad \text{(MHC-I)} \qquad (3)$$

where the interactions are given by the linear energy terms, $\beta_{ij} = \lambda\epsilon_i(a_i)\epsilon_j(a_j)$. Following the literature on fitness landscapes, we refer to these two types of nonlinearities as idiosyncratic and global epistasis, respectively [38, 41]. Under idiosyncratic epistasis, the energy effect of a mutation at a given position, $\Delta\Delta G$, depends on the detailed sequence content at other positions. Under global epistasis, it depends only on the energy of the wildtype sequence. In other words, the full energy can be written as a nonlinear function of an intermediate linear trait, here $\Delta G = L + \lambda L^2$ [34, 52].

These two types of energy landscapes are strongly different in model complexity. The idiosyncratic interaction term in Eq (2) has $20 \times 20$ energy parameters for the amino acid combinations at each pair of interacting peptide positions. The global epistasis model of Eq (3) requires only $20 \times \ell$ parameters for the linear term and a single additional parameter $\lambda$ measuring the strength of epistasis.

A common approach to infer epistasis from data is to compare the effect of a mutation on different sequence backgrounds, for example, by analyzing pairs of mutations at two positions in two different orders ($ab \to a'b \to a'b'$ and $ab \to ab' \to a'b'$). Here, given the limited availability of such pairs in our datasets, we map instead the deviations of partial data from the previously inferred linear model [53].

For MHC-I, we form data subsamples containing sequences with particular amino acid pairings in positions 2 and 9. For each subsample, we perform a linear regression, the slope and offset of which compare validation data with the linear model $L$ (dashed line). Fig 2A shows these regressions for a few subsamples of peptides binding HLA-A*02:01; all subsamples are shown in S2 Fig. We observe a systematic modulation of subsample slopes with linear energy $L$: deviations from the linear model increase from weak to strong binders (from left to right panel), but are similar for different subsamples in the same range of $L$. This pattern is consistent with global epistasis of the form $\Delta G(L) = L + \lambda L^2$, as described by Eq (3).

For MHC-II, here allele HLA-DRB1*01:01, we select subsamples with specific amino acid pairings in positions 4 and 6. In this case, the subsample slopes scatter strongly in the same

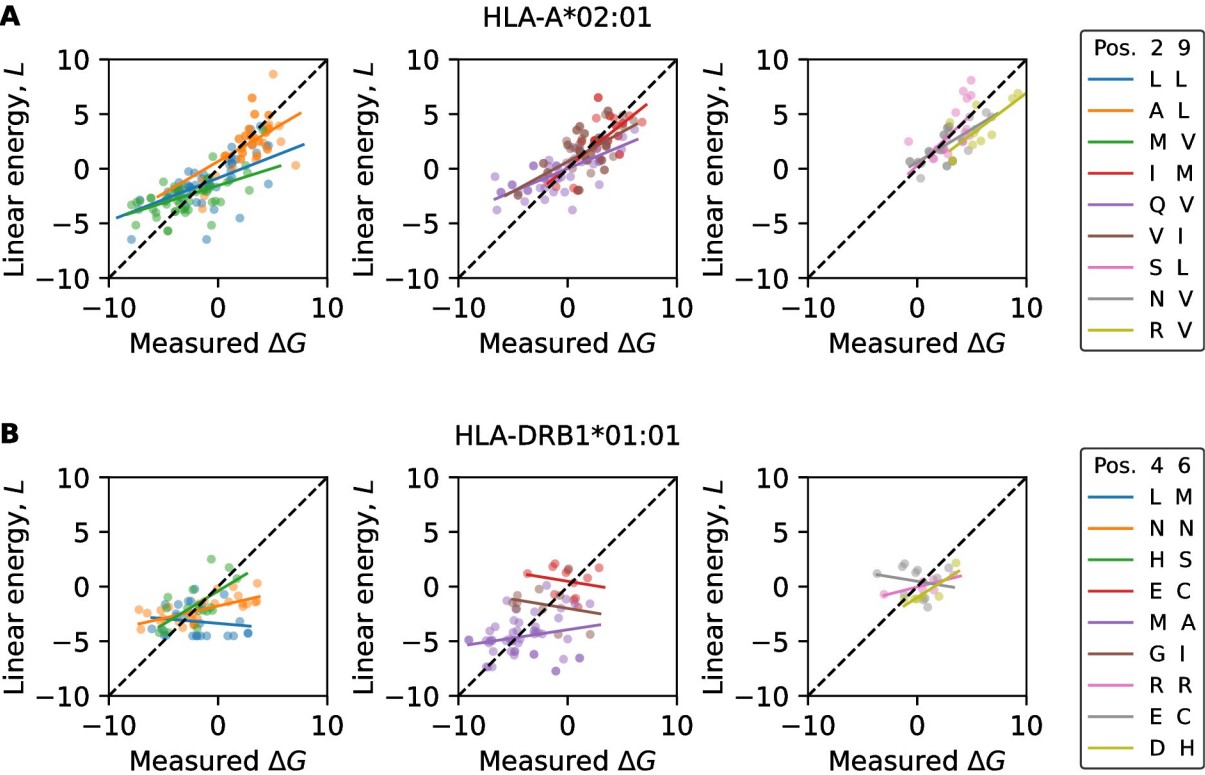

**Fig 2. Inference of epistasis.** Scatter plots of experimental $\Delta G$ values and the linear energy model $L$ for subsamples with given amino acid pairs in selected positions (max 50 random points are displayed). Top: HLA-A*02:01, selected positions 2 and 9; bottom: HLA-DRB1*01:01, selected positions 4 and 6. From left to right: subsamples containing stronger binders, marginal binders, and non-binders, respectively. Lines give linear regressions for each subsample. See S2 Fig for regression lines of all subsamples.

range of $L$ and there is no global trend from weak to strong binders (Fig 2B and S2 Fig). For example, the pairing Hystidine-Serine (H-S) shows a positive correlation between observed $\Delta G$ and $L$ in peptides with marginal binding, the pairing Leucine-Methionine (L-M) shows a negative correlation in the same range of $L$. Such disorder is seen ubiquitously, pointing to idiosyncratic epistasis depending on individual amino acid pairings, as described by Eq (2).

## Complexity of epistatic landscapes

The above analysis can be summarised using the difference between two errors: the MSE produced by a global function $\Delta G(L)$, fitted in the same $L$ region, compared to the MSE produced by a family of functions $\Delta G(L)$ fitted on subsamples of $L$ characterized by different amino-acid pairing in anchor positions (Fig 2 and S2 Fig, Material and methods). For the MHC-I allele HLA-A*02:01, we find an error difference $\Delta MSE = 0.6 \pm 0.4$, indicating that a global model is sufficient to describe the data and the marginal decrease in MSE of the subsample regressions does not justify the increase in model complexity. For the MHC-II allele HLA-DRB1*01:01, we obtain $\Delta MSE = 2.0 \pm 0.9$, suggesting that a more complex epistatic model is appropriate.

For comparison, we also analyze epistatic landscapes with random parameters. For idiosyncratic epistasis, we use energy landscapes of the form (2), where the parameters $\epsilon_i(a)$ and $\beta_{ij}(a, b)$ are Gaussian random variables with mean 0 and variance $\sigma_\epsilon^2$ and $\sigma_\beta^2$, respectively. For a given landscape, we evaluate the energy $\Delta G(\mathbf{a})$ for a sample of random peptides, subject to an additional measurement error of variance $\sigma_m^2 = 1.5$. The strength of epistasis is given by

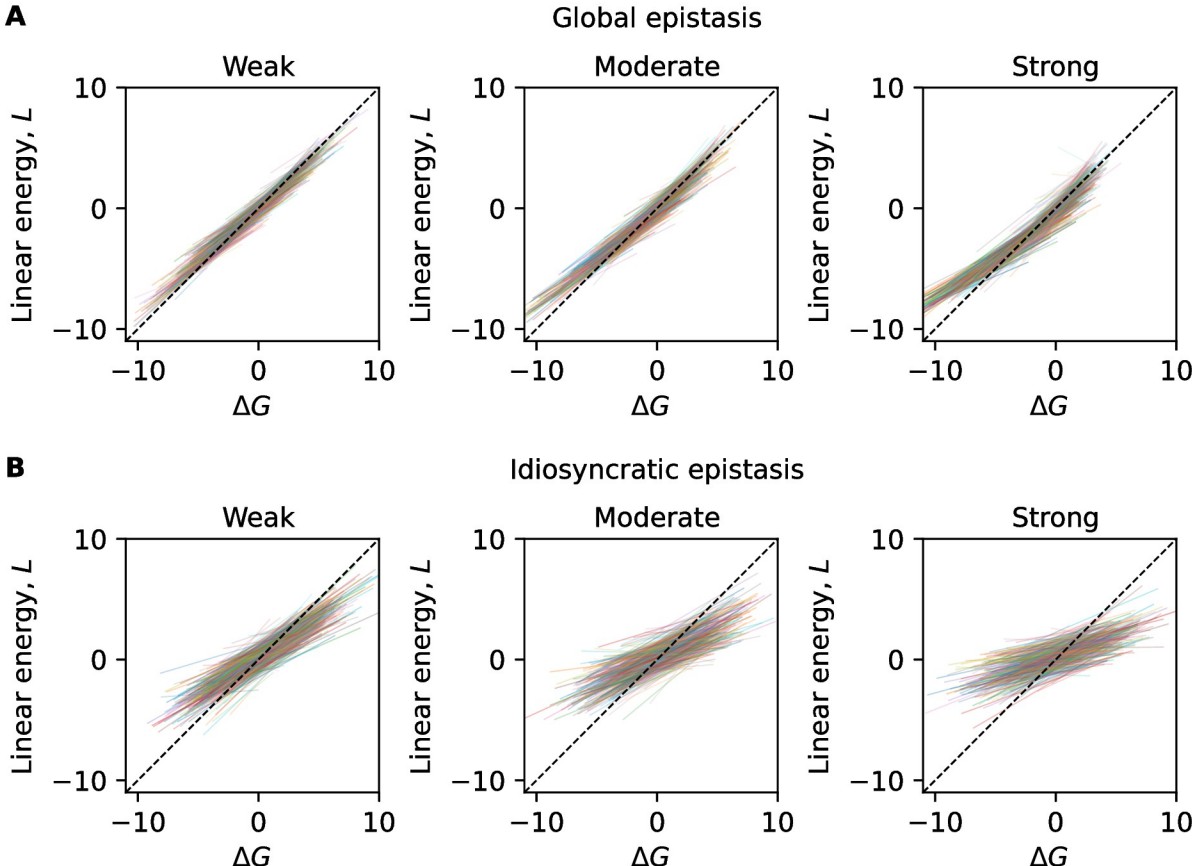

**Fig 3. Global and idiosyncratic epistasis in simulations.** Linear regression for subsamples of simulated data (one realization) according to the global epistasis model and the idiosyncratic epistasis model. A: From left to right, weak global epistasis with $\lambda = -0.01$, moderate with $\lambda = -0.04$ and strong $\lambda = -0.1$. The lines are color-coded based on the minimum L value spanned by the given subsample. B: From left to right, weak idiosyncratic epistasis with $\tilde{\lambda} = 0.3$, moderate with $\tilde{\lambda} = 1$ and strong $\tilde{\lambda} = 2$. The transparency of each regression line is proportional to the number of data points in the given subsample.

$\tilde{\lambda} = \sigma_\beta^2/\sigma_\epsilon^2$ (details of the simulations are given in Material and methods). With an appropriate strength ($\tilde{\lambda} = 2$), the simulated data generate a regression pattern strikingly similar to the empirical data of MHC-II (Fig 3B and S2 Fig), a similar error of the linear model (MSE = 6.2 ± 0.2), and a similar error difference between $\Delta$MSE = 2.3 ± 1.0, computed over 10 realizations. For global-epistasis landscapes of the form (3), the energy parameters $\epsilon_i(a)$ are again Gaussian random variables with variance $\sigma_\epsilon^2$, which is chosen according to the empirical values inferred for MHC-I alleles (S2 Table). The simulated data are drawn with a measurement error of variance $\sigma_m^2 = 1$. For intermediate strength of epistasis ($\lambda = -0.04$), the regression pattern is again similar to the empirical data (Fig 3A) and produces a similar error difference ($\Delta$MSE = 0.20 ± 0.1).

## Energy landscapes with global epistasis for MHC-I

The type of epistasis has immediate consequences for the learnability of landscape models from training data. For MHC-I, we can infer a global-epistasis model by regression: we learn the linear model $L(\mathbf{a})$ together with the parameter $\lambda$ of the nonlinear function $\Delta G(L)$. For the allele HLA-A*02:01, we infer an optimal quadratic model of the form (3) with $\lambda = -0.04$. This

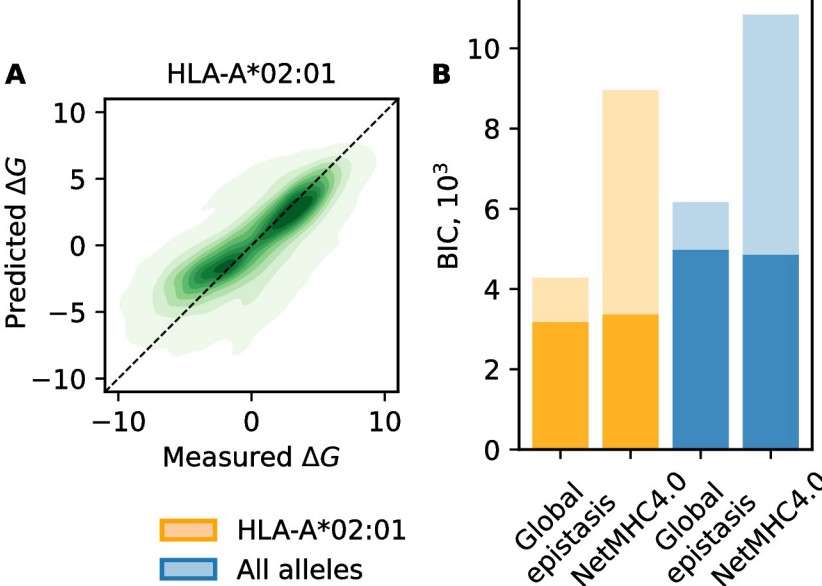

**Fig 4. Global-epistasis model for MHC-I.** A: Error plot comparing observed energies, $\Delta G$, and predictions of the optimal global-epistasis model ($\lambda = -0.04$) for HLA-A*02:01. To be compared with the linear model of Fig 1; see S1 Fig for other alleles. B: Model comparison of the global-epistasis model and NetMHC-4.0: error function (dark shading) and total Bayesian Information Criterion (BIC) score accounting for model complexity (full bars) for allele HLA-A*02*01 (orange) and aggregated over all 3 MHC-I alleles (blue), evaluated by blind testing (see text for details).

curvature is associated with moderate *increasing-return* epistasis: the global nonlinearity reduces the systematic error of the linear model in the strong-binding regime (Figs 1C and 4A), leading to a reduced mean square error (MSE = 4.3) compared to the linear model (MSE = 4.6) in the 10 fold cross-validation procedure. Adding a cubic nonlinearity to the function does not further reduce the MSE in a significant way. As shown below, the nonlinearity substantially increases the effects of mutations in strong binders. Very similar global energy landscapes are inferred for the other MHC-I alleles included in our analysis (S1 Fig).

Next, we compare the performance of the global epistasis model with the most widely used neural network model, NetMHC 4.0 [21], in both binding energy predictions and classification. As validation data for binding energy predictions, we use the set of all IEDB peptides that are not contained in the training data for NetMHC 4.0 (obtained from IEDB Analysis Resources [42, 43]). The set contains 465, 69 and 183 sequences for HLA*A-02:01, HLA*A-11:01 and HLA*B-07:02, respectively. These peptides are also excluded from the training of our model. The global-epistasis model for HLA*A-02:01 reaches a smaller error than NetMHC 4.0 (MSE = 6.8 compared to MSE = 7.2). The aggregated error computed on all MHC-I alleles included in the analysis reaches a slightly larger value (MSE = 6.9 compared to MSE = 6.8 for NetMHC 4.0), reflecting the reduced availability of training sequences (S3 Fig). To assess the classification performance, we collect sets of positive and negative binders from IEDB. We exclude sequences used for training and present in the referenced dataset [42, 43]. The performance metric is reported as the Area Under the Curve (AUC, see Material and methods). For HLA*A-02:01, we obtain a score of 0.837 for the global epistasis model and 0.843 using NetMHC 4.0. Scores for other alleles are detailed in S1 Table.

This level of performance is achieved with drastically lower model complexity: the global-epistasis model has $20 \times 9 + 1$ parameters per allele, significantly less than the number of

parameters required to account for all the pairwise interaction terms between each positions [17, 19]. In comparison, the network model employs $> 900$ parameters [21].

Factoring in the difference in model complexity by a Bayesian Information Criterion (Material and methods), the global-epistasis model strongly outperforms NetMHC 4.0 (Fig 4B). Hence, for the smooth energy landscapes of MHC-I alleles, neural network models are an overkill: they apply a complex multi-level learning procedure to an inference problem that can be solved by simple regression. In contrast, neural network models are an appropriate tool for the intrinsically more complex idiosyncratic energy landscapes of MHC-II. For HLA-DRB1*01:01, low-complexity models of the form (3) fail to map the energy variation (within-dataset MSE = 5.7), similar to the linear model shown in Fig 1C (cross-validated MSE = 5.9), while NetMHCII-2.3 still provides a faithful representation of the data (within-dataset MSE = 1.6).

## Mutational effect spectra in epistatic landscapes

We can use the energy data of MHC-I and MHC-II for a comparative analysis of mutational effects. In Fig 5A and 5B, we show the distribution of energy changes, $\Delta\Delta G$, plotted separately for mutations at anchor and non-anchor positions and for ancestral sequences in two affinity windows: typical binders ($\Delta G_{\text{anc}} \approx -1$) and strong binders ($\Delta G_{\text{anc}} \approx -5$). For MHC-I, we observe homogeneously upscaled effects for mutations of strong binders, reflecting the larger slope of the global epistasis function $\Delta G(L)$ (Fig 5A). This upscaling increases mean and variance of magnitude effects, $E(|\Delta\Delta G|)$ and $\text{Var}(|\Delta\Delta G|)$, in non-anchor and, at larger amplitudes, in anchor positions (inserts in Fig 5). For MHC-II, the mean effect $E(|\Delta\Delta G|)$ is also enhanced in strong binders (Fig 5B). This is the known global pattern of idiosyncratic epistasis [41]: strong binders have a higher fraction of affinity-increasing bonds (i.e., amino acid pairings for which $\beta_{ij}(a_i, a_j) < 0$), generating an additional energy increase for mutations breaking these bonds. However, the variance of magnitude effects is largely independent of $\Delta G_{\text{anc}}$, and the effect distributions are similar in anchor and non-anchor positions. This is again consistent with strong idiosyncratic epistasis: large mutational effects at non-anchor positions can be generated by energy bonds to anchor positions.

To quantify idiosyncratic epistasis directly, we record mutational effect spectra generated by background sequence variation from calibrated and random energy models for MHC-II (these spectra are undersampled in the data). For a given mutation $a \to a'$ of a peptide with given energy $\Delta G_{\text{anc}}$, we resample the ancestral background sequence at the other positions subject to the constraint of maintaining $\Delta G_{\text{anc}}$; the resulting effect spectrum is then shifted to mean 0. In S4 Fig, we plot these spectra for a sample of mutations in typical binders ($\Delta G_{\text{anc}} \approx -1$) and strong binders ($\Delta G_{\text{anc}} \approx -5$), together with the corresponding sample-averaged distributions. The rms effect given by these distributions measures idiosyncratic deviations from global epistasis. This effect is sizeable for typical and strong binders and similar to the global shift in $E(|\Delta\Delta G|)$ between the two binding regimes (Fig 5B and S4 Fig). Hence, in MHC-II, the energy effect of a given mutation depends strongly on the specific amino acid pairings of the ancestral peptide.

## Immune escape evolution

The energy statistics of peptide-MHC binding affect the escape evolution of antigens from T cell immune recognition. In both MHC classes, even strong-binding antigenic epitopes can effectively suppress presentation by large-effect mutations ($\Delta\Delta G \gtrsim 5$, shaded areas in Fig 5A and 5B). For MHC-I, large-effect mutations occur in anchor positions, while most mutations in non-anchor positions maintain presentation. This reflects the modularity of presentation

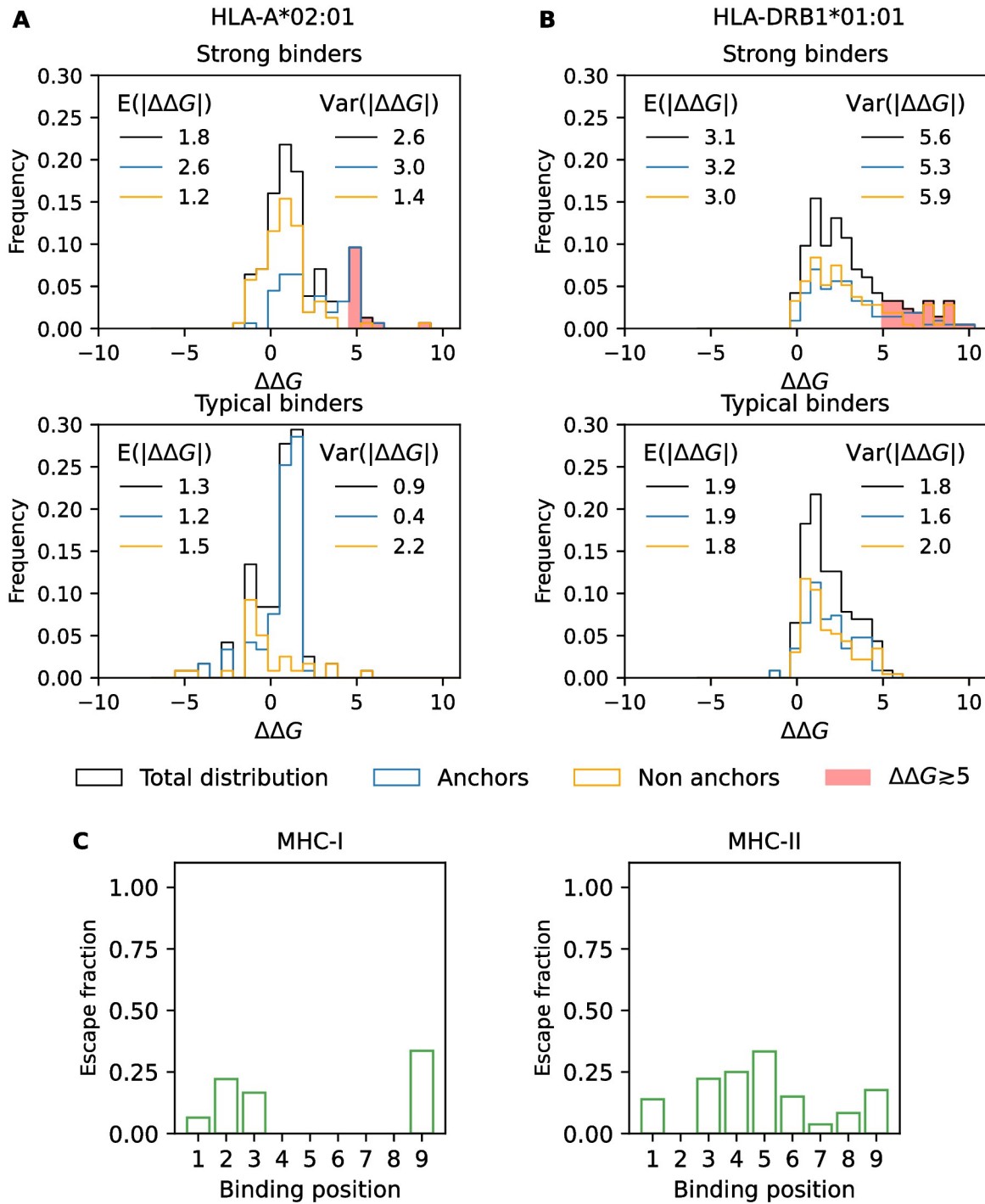

**Fig 5. Mutational effect distributions.** A-B: Distribution of energy changes, $\Delta\Delta G$, for mutations from ancestral peptides in different energy windows: strong binders ($\Delta G \approx -5$, top) and typical binders ($\Delta G \approx -1$). The total distribution (black) is shown together with the components from anchor positions (blue) and non-anchor positions (orange). For strong binders, red shading marks escape mutations leading to loss of presentation ($\Delta\Delta G \gtrsim 5$). Inserts give mean and variance of magnitude effects $|\Delta\Delta G|$. A: MHC-I, allele HLA-A*02:01 (anchor positions 2 and 9). B: MHC-II, allele HLA-DRB1*01:01 (anchor positions 1, 4, 6 and 9). C: Distribution of escape from MHC binding in HIV. Position-dependent fraction of loss-of-binding mutations inferred from a set of HIV epitopes presented by MHC-I or MHC-II; see text and S9 Table.

and T cell recognition encoded in peptide sequences. For MHC-II, escape mutations acting on presentation are broadly distributed across anchor and non-anchor positions. The overall target for this type of escape evolution is larger for MHC-II: 21% of all mutated strong binders have $\Delta\Delta G \geq 5$, compared to 12% for MHC-I. This suggests that the idiosyncrasy of peptide-MHC-II binding facilitates the escape evolution of antigens by generating large epistatic mutation steps.

To test our analysis, we compare mutations of MHC ligands in HIV-derived epitopes for multiple alleles of MHC-I and MHC-II. From IEDB, we collect 1299 pairs of HIV epitopes that differ in a single amino acid affecting MHC binding (Material and methods). In a subset of these pairs, the epitopes have different binding annotations (positive, negative), indicating a likely escape mutation from MHC binding. A limitation of this inference is that the two epitopes are, in general, from viral probes of different individuals. In Fig 5C, we plot the position-specific fraction of inferred escape mutations. For MHC-I, the majority of escape mutations are at positions 2 and 9. MHC-II shows a higher overall fraction of escape mutations and a broader distribution of positions. This is consistent with our inference of mutational targets and suggests that HIV can escape T cell immune response by suppressing presentation.

## Discussion

Peptide-MHC binding interactions have strong, class-specific nonlinearities. For MHC-I, these interactions can be described by an energy landscape with global epistasis, $\Delta G(L)$, where the intermediate trait $L$ is additive in the peptide sequence. The curvature of the landscape increases in slope at strong binding. This enhances the energy discrimination of presented peptides and increases selection on affinity-decreasing escape mutations in pathogen evolution. Moreover, the recognition chain of class I is modular: strong MHC binding energy effects governing peptide presentation are predominantly determined by the anchor positions 2 and 9, while non-anchor positions encode the extracellular interactions of presented peptides with T cell receptors. For MHC-II, we find rugged energy landscapes with idiosyncratic epistasis, describing cooperative binding that depends on the amino acid content at multiple peptide binding positions. Mutational effects at a given binding position depend strongly on the peptide sequence background, and there is no modularity: weak and strong effects are found at anchor and non-anchor positions.

These differences in epistasis between energy landscapes are in tune with distinct biophysical binding mechanisms. MHC-I molecules bind short peptides in a fixed contact map; global epistasis of the binding energy is consistent with mutual reinforcement of binding between the anchor positions that is mediated by the intermediate chain. In contrast, MHC-II binds longer peptides with variable contact maps [16], suggesting that idiosyncratic epistasis can partly be attributed to changes in the core contact points and in the peptide configuration in between these points.

The type of epistasis also governs the efficient learning of energy landscapes from data. For MHC-I, we derive a simple energy matrix model that outperforms state-of-the-art machine-learning models [21] in BIC score. This model can be inferred by a joint regression procedure for the linear trait $L(\mathbf{a})$ and for the nonlinear function $\Delta G(L)$. Such regression methods fail for MHC-II, while neural networks still produce a faithful energy landscape [13, 28]. The difference in learnabilty is not surprising, given the strongly different complexity of global and idiosyncratic energy models. This compression can be understood as a dimensionality reduction in phenotype space. In the idiosyncratic case, sequence variation affects not only the focal trait, here the binding energy $\Delta G$, but also a number of co-varying traits $A, B, \ldots$, including the spatial configuration of the interaction surface and allosteric interactions. A given mutation $a \rightarrow$

$a'$ on two different backgrounds $b$, $b'$ has effects $\Delta\Delta G_b = \Delta\Delta G(\Delta G_{ab}, A_{ab}, B_{ab}, \ldots)$ and $\Delta\Delta G_{b'} = \Delta\Delta G(\Delta G_{ab'}, A_{ab'}, B_{ab'}, \ldots)$. These effects differ because they probe the slope of the energy landscape at different values of the co-varying traits, even if they start from similar energies, $\Delta G_{ab} \approx \Delta G_{ab'}$. Under global epistasis, the ensemble of traits collapses to a single relevant trait $\Delta G$. This implies $\Delta\Delta G_b = \Delta\Delta G(\Delta G_{ab})$ and $\Delta\Delta G_{b'} = \Delta\Delta G(\Delta G_{ab'})$; the only remaining source of epistasis for the mutation $a \rightarrow a'$ is the energy effect of the background mutation $b \rightarrow b'$. The (lack of) collapse from idiosyncratic to global epistasis can be evaluated in a given training dataset and serve as a diagnostic for the appropriate learning procedure. The main limitation of energy model inference remains the availability of binding data for less common MHC alleles.

In summary, we have highlighted the roles of epistasis in the peptide-MHC energy landscapes for T cell immunity. The type of epistasis is linked to the underlying biophysical mechanism, governs the learnabilty of landscape models from data, and shapes the evolution of binding agents (here, antigenic peptides). Accounting explictly for global and position-specific epistasis may provide an avenue towards better MHC-binding models, which is critical to efficiently scan pathogen and cancer genomes for likely T cell antigens. The broad picture of trait-based epistasis is expected to be applicable to other protein-protein interaction landscapes.

## Material and methods

### Dataset curation

The datasets for both MHC-I and MHC-II are sourced from the IEDB database [42] (version April 2023 for HLA-A*02:01, May 2023 for all other alleles), reporting measured IC50 values in nM. The experimental methods involve only in vitro binding assays, no mass spectrometry measurements are selected. This dataset contains quantitative and qualitative measurements. We first discard peptides containing non-canonical amino acids. Several sequences contain multiple measurements, which are curated in the following way. For a set of quantitative $\Delta G$, we average $\Delta G$ values over multiple measurements relative to the same sequence from training and test data. We accept values within one standard deviation from the mean and discard others. For sets of qualitative measurements ($\Delta G > \Delta G_{\mathrm{thr}}$), we select the minimum $\Delta G_{\mathrm{thr}}$. We generate cross-validation training and test sets, taking care not to include the same sequence in both sets at the same time.

### Extraction of MHC-II cores

From IEDB, we collect sequences of 15-mers along with their binding affinities. However, our binding model assumes that, in good approximation, the binding involves mainly their core of 9 aa that directly binds to the complex. This assumption allows us to extract the binding core and infer a $20 \times 9$ energy matrix, in the same fashion as MHC-I. An estimation of the core for longer sequences in input is provided by NetMHCII-2.3 [28]. The extracted 9-mers are paired with the experimental measures provided by IEDB and used for the inference of the linear model.

### Model optimisation

The energy matrix and coefficients are trained and tested using a 10-fold cross-validation approach: we randomly divide the data in 10 subsets, 9 of which are used for training and one for testing. We optimize the error function

$$n \times \mathrm{MSE} = \sum_{i=1}^{n} (\Delta G_{i,\mathrm{model}} - \Delta G_{i,\mathrm{data}})^2,\tag{4}$$

where $n$ is the number of sequences within the test set.

The function in Eq (4) is adapted for inequalities present in the datasets. For qualitative data with $\Delta G > \Delta G_{\text{thr}}$, we use the error 0 if $\Delta G_{\text{model}} > \Delta G_{\text{thr}}$ and $(\Delta G_{\text{model}} - \Delta G_{\text{thr}})^2$ otherwise. For the linear model, we start from a matrix of randomly generated $\epsilon$, each drawn by a Gaussian distribution with $\sigma^2 = 1/180$. In order to speed up the minimization, we add another degree of freedom, $\epsilon_0$, which can be inserted back into a $20 \times 9$ energy matrix without changes in the outcome of the minimization. The results of the linear optimization are used as a starting point to optimize the global epistasis model. For this reason, we use the same training partitions to avoid biased results. The minimized global model is of the form $\Delta G = k + aL' + bL'^2$, where 2 degrees of freedom are added in order to speed up the computation. Via the transformation for $L = k + aL'$, we can express $\Delta G$ as $L + \lambda L^2$, where $\lambda$ has units of $1/k_bT$. With this transformation, we bring the essential parameters to 181. The linear and epistatic energy matrices trained on the full datasets of all alleles studied are given in S2–S8 Tables.

## Complexity estimates

For the analysis of Fig 2, we divide linear predictions into bins of the linear trait $L$. For each bin, we fit the data $\Delta G$ to an average trait $L'_{\text{ave}} = aL + b$ and to subset-specific forms $L'_p = a_p L + b_p$ for specific combinations of alleles at anchor positions. The difference between the corresponding errors, $\Delta\text{MSE} = \text{MSE}_{\text{ave}} - \text{MSE}_p$, serves as an estimate for the added value of models with idiosyncratic epistasis, as given by Eq (2).

For estimating the complexity of the given MHC class, we compute $\Delta\text{MSE}$ in bins of $L$ with size 2, spanning $L$ values from -11 to 15 for both MHC classes (other bin sizes give broadly consistent results). In each bin, we evaluate $\Delta\text{MSE} = \text{MSE}_{\text{ave}} - \text{MSE}_p$ over a set of $p$ subsamples with specific allele combinations, including only subsamples with more than 2 data points.

The $\Delta\text{MSE}$ values from simulations are derived from randomly generated sets of 20,000 peptides. The values reported in the main text are obtained by averaging over 10 realizations, and the standard deviation is computed across bins over these realizations. The range of $L$ spanned in the simulated data is determined based on the minimum and maximum values observed in the specific realization under consideration.

## Simulations of random energy models

The idiosyncratic model used for simulations incorporates 4 pairwise interactions among positions 2 and 6, 2 and 9, 4 and 6, 4 and 9. We draw randomly the energy and interaction matrices from a Gaussian distribution centered at 0. The distribution of experimental data exhibits a total variance $\sigma^2 = \sigma_m^2 + \sigma_\epsilon^2 + \sigma_\beta^2 = 8.7$. From data relative to HLA-DRB1*01:01, we estimate $\sigma_m^2 = 1.5$ as a measure of the experimental noise. We can then choose $\sigma_\beta^2$ and $\sigma_\epsilon^2$ by fixing their ratio, defined as $\tilde{\lambda}$. We simulate 3 different regimes of epistatic strength: strong ($\tilde{\lambda} = 2$), moderate ($\tilde{\lambda} = 1$) and weak ($\tilde{\lambda} = 0.3$).

In the simulations of the global epistasis model, each column of the energy matrix is inferred from a Gaussian distribution with a zero mean and a standard deviation corresponding to the respective column of the linear energy matrix (S2 Table). We consider 3 different regimes of epistatic strength: weak ($\lambda = -0.01$), moderate ($\lambda = -0.04$) and strong ($\lambda = -0.1$). The systematic noise, denoted as $\sigma_m$, is introduced from a Gaussian distribution with a variance of 1, computed from data for HLA-A*02:01.

For both classes, we perform 10 separate realizations for each $\tilde{\lambda}$ and $\lambda$, characterized by different random sets of training sequences and initial conditions for $\epsilon$. A linear model is optimized and tested on random test sets. The partial regressions, performed on similar $L$ ranges, are used to estimate the complexity of each landscape.

## Classification score

The classification score is computed on a set of positive and negative binders, sourced from IEDB on March, 4th 2024 for all the considered alleles. After curations, the validation sets consist of 2488 positive and 1497 negative binders for HLA*A-02:01, 391 positive and 184 negative binders for HLA*A-11:01 and 382 positive and 505 negative binders for HLA*B-07:02.

We use the curated sets to construct the Receiver Operating Characteristic (ROC) curves, illustrating the relationship between *sensitivity*, representing the ratio of true positives to the total number of positives, and 1−*specificity*, which denotes the ratio of false positives to the total negatives [17, 54]. We define positives and negatives based on the binding probability defined below:

$$p = \frac{1}{1 + e^{\Delta G}}. \tag{5}$$

We calculate these metrics by varying the thresholds that distinguish positives from negatives and computing the corresponding variables. The area under this curve provides the AUC score, which is presented for the relevant alleles in S1 Table.

## Model comparison

We use the BIC score [55],

$$\text{BIC} = n \times \text{MSE} + k \log(n), \tag{6}$$

where $n$ is the number of sequences in the test set and $k$ the number of parameters, respectively. For NetMHC 4.0, we compute $k = k_h + k_{out}$, in which $k_h = (n_{in} + 1) \times n_h$, where $n_{in}$ and $n_h$ are the number of inputs and hidden neurons. Similarly, $k_{out} = (n_h + 1) \times n_{out}$, where $n_{out}$ is the number of output neurons [56]. With the architecture information provided in reference [21], $n_{in} = 180$, $n_h = 5$ and $n_{out} = 1$, we obtain $k$ is 911. The total BIC is reported in Fig 4B.

## Mutational effect spectra

The mutational effect in Fig 5 is computed by collecting all pairs of single mutants available in the data. Specifically, we collect 156 pairs in the strong binders regime and 119 for typical binders regime for HLA-A*02:01. For HLA-DRB1*01:01, we collect 214 and 230 sequence pairs for strong and typical binders regimes, respectively. For each energy window, both quantitative and qualitative measurements are collected. For mutations with qualitative data, $\Delta G_{wt} > \Delta G_{thr}$, we define $\Delta\Delta G = 0$ if $\Delta G_{mut} > \Delta G_{thr}$ and $\Delta\Delta G = \Delta G_{mut} - \Delta G_{thr}$ otherwise; mutations with a qualitative $\Delta G_{mut}$ are treated in an analogous way. The spectra of background mutations in S4 Fig are computed based on 50 single mutants presenting the same mutation at the chosen position. The ancestral peptides are selected in the same energy window for strong and typical binders, as above. For both the idiosyncratic model and NetMHCII 2.3, positions 4 and 6 are chosen as main anchors.

## HIV mutations

HIV-derived sequences have been downloaded from IEDB [42] on May 29, 2024. We collect 9-mers for MHC-I and sequences of length ≤ 20 aa for MHC-II. We restrict the analysis to alleles with >10 available sequences and compute the binding core of MHC-II sequences with NetMHCII-2.3. We extract pairs of peptides that differ by one amino acid and infer escape mutations from a change in binding annotation between the two peptides (Positive vs.

Negative). We record the fraction of inferred escape mutations for each position of the binding core (Fig 5C).

## Supporting information

**S1 Fig. Error plots for HLA-A\*11:01 and HLA-B\*07:02.** Error plot compares $\Delta G$ from validation data and predictions of the linear model (in red), $L$ (dashed line, left). On the right (in green), validation data for $\Delta G$ are compared to the global epistasis model (for both alleles, $\lambda = -0.04$). The error is computed across 10 cross-validation test sets (contours give densities above 0.01). A: HLA-A\*11:01 (Linear model MSE = 5.0, Global model MSE = 4.4) B: HLA-B\*07:02 (Linear model MSE = 6.3, Global model MSE = 5.6).
(PDF)

**S2 Fig. Inference of epistasis on full datasets.** Linear regressions summarising errors of L with respect to $\Delta G$, for each subsample available for HLA-A\*02:01 (194 linear fits, on the left) and HLA-DRB1\*01:01 (351 linear fits, on the right). Each line is color-coded depending on the start of the L region spanned by the given subsample. The transparency of each regression line is proportional to the number of data points in the given subsample.
(PDF)

**S3 Fig. Blind set validation for MHC-I alleles.** Error plot compares observed energies contained in the blind set, $\Delta G$, and predictions of the global epistasis model with $\lambda = -0.04$ for all the tested alleles (A, on top) and NetMHC 4.0 (B, bottom). Contours give densities above 0.01. The blind set contains 465 sequences for HLA-A\*02:01, 69 for HLA-A\*11:01 and 183 for HLA-B\*07:02.
(PDF)

**S4 Fig. Idiosyncratic deviations from global epistasis.** Background mutation spectra, shifted to mean 0, for samples of mutations in allele HLA-DRB1\*01:01 for strong binders ($\Delta G \approx -5$, top) and typical binders ($\Delta G \approx -1$, bottom). A: Spectra computed from the random idiosyncratic model with $\tilde{\lambda} = 2$; B: spectra computed with NetMHCII-2.3.
(PDF)

**S1 Table. Classification comparison.** AUC score computed between the global epistasis model and NetMHC 4.0 for all alleles included in this work. The allele name is reported in the first column, the second column contains the score obtained for the Global epistasis model in Eq (3), and the third column the score obtained using NetMHC 4.0.
(CSV)

**S2 Table. Position weight matrix for the linear model, $L(a)$, in HLA-A\*02:01.** Linear energy matrix inferred from experimental data for HLA-A\*02:01. The first column reports the amino acid name. The following columns report the amino acid-dependent contribution to the total energy for each peptide position.
(CSV)

**S3 Table. Position weight matrix for the global epistasis model, with $\lambda = -0.04$ in HLA-A\*02:01.** Energy matrix inferred according to the global epistasis model. The first column reports the amino acid name. The following columns report the amino acid-dependent contribution to the linear energy term for each peptide position.
(CSV)

**S4 Table. Position weight matrix for the linear model, $L(a)$, in HLA-A\*11:01.** Linear energy matrix inferred from experimental data for HLA-A\*11:01. The first column reports the amino

acid name. The following columns report the amino acid-dependent contribution to the total energy for each peptide position.
(CSV)

**S5 Table. Position weight matrix for the global epistasis model, with λ = − 0.04 in HLA-A\*11:01.** Energy matrix inferred according to the global epistasis model. The first column reports the amino acid name. The following columns report the amino acid-dependent contribution to the linear energy term for each peptide position.
(CSV)

**S6 Table. Position weight matrix for the linear model, $L(a)$, in HLA-B\*07:02.** Linear energy matrix inferred from experimental data for HLA-B\*07:02. The first column reports the amino acid name. The following columns report the amino acid-dependent contribution to the total energy for each peptide position.
(CSV)

**S7 Table. Position weight matrix for the global epistasis model, with λ = − 0.04 in HLA-B\*07:02.** Energy matrix inferred according to the global epistasis model. The first column reports the amino acid name. The following columns report the amino acid-dependent contribution to the linear energy term for each peptide position.
(CSV)

**S8 Table. Position weight matrix for the linear model, $L(a)$, in HLA-DRB1\*01:01.** Linear energy matrix inferred from experimental data for HLA-DRB1\*01:01. The first column reports the amino acid name. The following columns report the amino acid-dependent contribution to the total energy for each peptide core position.
(CSV)

**S9 Table. MHC-I and MHC-II alleles and number of HIV-derived sequence pairs.** Total single mutations for the MHC-I and MHC-II alleles. The allele names are listed in the first column: MHC-I alleles are collected from HLA-A\*02:01 to HLA-B\*54:01, and the remaining rows contain all MHC-II alleles. The total number of single mutations is given in the second column.
(CSV)

## Acknowledgments

We thank Leon Seeger and Denny Trimcev for discussions and comments on the manuscript.

## Author Contributions

**Conceptualization:** Laura Collesano, Marta Łuksza, Michael Lässig.

**Data curation:** Laura Collesano.

**Formal analysis:** Laura Collesano.

**Funding acquisition:** Michael Lässig.

**Investigation:** Laura Collesano.

**Methodology:** Laura Collesano, Michael Lässig.

**Project administration:** Michael Lässig.

**Resources:** Michael Lässig.

**Software:** Laura Collesano.

**Supervision:** Marta Łuksza, Michael Lässig.

**Validation:** Laura Collesano, Michael Lässig.

**Visualization:** Laura Collesano.

**Writing – original draft:** Laura Collesano, Michael Lässig.

**Writing – review & editing:** Laura Collesano, Marta Łuksza, Michael Lässig.

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
