## [Decision Letter · Decision Letter 0]

14 May 2024

Dear Ms. Collesano,

Thank you very much for submitting your manuscript "Energy landscapes of peptide-MHC binding" for consideration at PLOS Computational Biology. As with all papers reviewed by the journal, your manuscript was reviewed by members of the editorial board and by several independent reviewers. The reviewers appreciated the attention to an important topic. Based on the reviews, we are likely to accept this manuscript for publication, providing that you modify the manuscript according to the review recommendations.

Sincerely,

Piero Fariselli

Academic Editor

PLOS Computational Biology

Nir Ben-Tal

Section Editor

PLOS Computational Biology

Reviewer's Responses to Questions

**Comments to the Authors:**

Reviewer #1: The authors have attempted to design computationally sequence-dependent energy landscapes of MHC-peptide binding encode class-specific nonlinearities (epistasis). MHC-I has a smooth landscape with global epistasis; the binding energy is a simple deformation of an underlying linear trait. This form of epistasis enhances the discrimination between strong-binding peptides. This study offers a fascinating insight into the intricate world of peptide-MHC binding and its role in immune recognition. The distinction between MHC-I and MHC-II energy landscapes sheds light on their respective complexities. The research underscores the significance of sequence-dependent energy landscapes in peptide-MHC binding and their implications for T cell immunity. It's intriguing to see how MHC-I exhibits a smoother energy landscape with global epistasis, while MHC-II displays a more rugged landscape with idiosyncratic epistasis. The utilization of datasets from the Immune Epitope Database adds credibility to the findings, ensuring a robust analysis of peptide-MHC interactions. The study's approach in inferring linear energy models and exploring epistatic landscapes provides valuable insights into the mechanics of immune recognition. The comparison between linear and global-epistasis models elucidates the suitability of different modeling approaches for MHC-I alleles. The performance evaluation against NetMHC 4.0 highlights the efficacy of the proposed global-epistasis model, particularly for MHC-I alleles. Having said that, the paper could still be a good reference for similar studies, and thus, I recommend its publication after the author addresses all the major and minor concerns satisfactorily.

Major concerns

Abstract: The abstract provides a clear overview of the study; however, it would benefit from a brief mention of the significance of the findings and their potential implications in the field of immunology.

Introduction: While the introduction sets the stage for the study, it lacks a clear statement of the research gap or question addressed by the study. Providing a more explicit statement of the problem would enhance the introduction.

Methods: The study's methodology for dataset curation ensures the reliability of the findings by incorporating quality control measures. It's impressive to see how the research integrates experimental data with computational modeling to unravel the complexities of peptide-MHC interactions. The comparison between linear and epistatic models highlights the importance of considering higher-order interactions in accurately modeling peptide-MHC binding. Overall, this study significantly contributes to our understanding of immune recognition mechanisms and has implications for the development of novel therapeutic interventions. The study's comprehensive approach, encompassing experimental data analysis and computational modeling, ensures a thorough exploration of peptide-MHC binding dynamics.

The methods section is detailed and well-structured, providing clarity on data collection, analysis, and model optimization. However, including information on potential limitations of the methods or assumptions made during the analysis would improve the section's comprehensiveness.

Results: The results section presents key findings regarding the energy landscapes of MHC-peptide binding. Adding more contextualization of the results within the broader research landscape would enhance the section's readability.

Figures and Tables: The figures and tables are informative and well-designed. However, providing more detailed figure legends and ensuring consistency in labeling throughout the manuscript would improve clarity.

Discussion: The discussion should not only summarize the findings but also provide insights into their significance and potential impact on the field. Additionally, discussing any unexpected results or limitations of the study would add depth to the discussion.

The analysis of mutational effect spectra offers a deeper understanding of the impact of mutations on peptide-MHC binding, especially in the context of escape evolution. The observation of modularity in presentation and T cell recognition underscores the complexity of immune evasion strategies employed by pathogens. The study's methodology, including dataset curation and model optimization, ensures rigor in the analysis of peptide-MHC interactions. It's noteworthy how the study addresses the challenge of complexity in modeling peptide-MHC interactions, particularly in the case of MHC-II. The simulations of random energy models provide a useful tool for understanding the underlying mechanisms driving peptide-MHC binding.

The research contributes to our understanding of immune escape evolution by highlighting the distinct mechanisms employed by MHC-I and MHC-II. The findings regarding the distribution of large-effect mutations offer valuable insights into the vulnerabilities of antigenic epitopes to escape mutations. The study's focus on the interplay between sequence-dependent energy landscapes and immune recognition provides a holistic view of the immune system's functioning. The exploration of different regimes of epistasis enhances our understanding of the adaptive strategies employed by pathogens to evade immune detection. The analysis of energy statistics underscores the importance of considering both strong-binding antigenic epitopes and typical binders in studying immune escape evolution.

Language: The manuscript generally adheres to academic writing standards. However, some sentences are overly complex and could be simplified for better readability. Additionally, ensure consistency in terminology and avoid jargon when possible.

Grammatical errors: The manuscript contains several grammatical errors and awkward sentence constructions. A thorough proofreading and editing process is necessary to correct these issues and enhance the clarity of the text.

Clarity of Presentation: The manuscript would benefit from clearer transitions between sections to guide the reader through the logical flow of the study. Additionally, ensuring that each paragraph focuses on a single idea would improve readability.

Citation: Ensure that all references are cited correctly and consistently throughout the manuscript, following the appropriate citation style guidelines.

Statistical Analysis: Provide more detailed information on the statistical methods used, including assumptions made and measures of uncertainty or variability.

Future Directions: Conclude the manuscript with a brief discussion of potential future research directions or avenues for further exploration in this area. The conclusion should reiterate the main findings of the study and emphasize their importance in advancing knowledge in the field of immunology.

Graphical Abstract: Consider creating a graphical abstract to visually summarize the key findings of the study and attract readers' attention.

Impact Statement: Include a brief impact statement highlighting the significance of the study findings and their potential implications for research, clinical practice, or policy development.

Missing references: https://doi.org/10.1038/s42256-022-00459-7;

Final Remarks: Final remarks summarizing the key points addressed in the review and expressing confidence in the manuscript's potential for publication with revisions. I would like to see the revised version of this manuscript.

Reviewer #2: Summary

In this paper, the authors fit models of varying simplicity to recapitulate data on peptide binding to human HLA molecules. This task is currently performed by NETMHCpan which uses a neural network trained on the same data as the authors use in this paper. However, this paper sheds new light on peptide binding to MHC. The main result is that a simple multiplicative quadratic model (global epistasis) can fit the data for peptide binding to MHC I molecules as well as NETMHCpan with far fewer parameters; i.e., for MHC I neural nets are an overkill. The authors also show that for MHC II molecules more complex models are required and the use of neural nets for obtaining accurate models is justified. These results make intuitive sense as the MHC I groove is closed at the ends with strong dependence of peptide binding on the two key anchor positions (2 and 9). On the other hand, the MHC II groove is open at the ends and the binding register can shift for different peptides binding to the same class II allele. But, to my knowledge, the authors are the first to demonstrate this quantitatively. Therefore, I recommend publication of the paper in PLOSCB after the reviewers have a chance to revise the manuscript to address the points noted below.

Specific points:

1] Why is �S a measure for how good a linear model would be? For example, �S could be large for class I molecules because the value at the 2 and 9 positions are very high. I could not understand why this pertains to how good a linear model will be.

2] There is a syntax problem on line 451.

3] In lines 139-144 it is first noted that the likely factors for bad agreement for MHC II with the linear model are variation in binding cores and additional interactions from flanking regions. But then it is noted that epistasis is the cause. Some clarity is required here.

4] For class I molecules, would the fit be of poorer quality than that obtained using Eq. 3 if the interactions between the amino acids of the peptide and the HLA molecule were taken to be those in the Miyazawa-Jerningan (MJ) matrix that is used to model protein folding?

5] What is meant by “original sequence” on line 163?

6] I suggest that Fig. S3 be moved to main text as there is a whole section describing it in the main text and it is informative.

7] The acronym BIC should be defined in Fig. 3 – the acronym is never defined in the main text where it is written as Bayesian information criterion.

8] Is there sufficient data for escape mutations due to loss of MHC presentation during HIV infection to see if they are largely concentrated at anchor positions for MHC I and distributed all over for MHC II? This would be consistent with the authors’ findings, and if so, the paper would be strengthened.

9] Define the meaning of physiological binding threshold in line 98 better as it sets the scale for �G.

**Have the authors made all data and (if applicable) computational code underlying the findings in their manuscript fully available?**

Reviewer #1: None

Reviewer #2: None

PLOS authors have the option to publish the peer review history of their article (what does this mean?). If published, this will include your full peer review and any attached files.

Reviewer #1: No

Reviewer #2: No

Figure Files:

Data Requirements:

Reproducibility:

References:

---

## [Decision Letter · Decision Letter 1]

31 Jul 2024

Dear Ms. Collesano,

We are pleased to inform you that your manuscript 'Energy landscapes of peptide-MHC binding' has been provisionally accepted for publication in PLOS Computational Biology.

Best regards,

Piero Fariselli

Academic Editor

PLOS Computational Biology

Nir Ben-Tal

Section Editor

PLOS Computational Biology

Reviewer's Responses to Questions

**Comments to the Authors:**

Reviewer #1: The revision is satisfactory, it is acceptable in its current form

Reviewer #2: The authors have addressed all my previous points adequately.

**Have the authors made all data and (if applicable) computational code underlying the findings in their manuscript fully available?**

Reviewer #1: Yes

Reviewer #2: None

PLOS authors have the option to publish the peer review history of their article (what does this mean?). If published, this will include your full peer review and any attached files.

Reviewer #1: No

Reviewer #2: No

---

## [Editor Report · Acceptance letter]

27 Aug 2024

PCOMPBIOL-D-24-00472R1 

Energy landscapes of peptide-MHC binding

Dear Dr Collesano,

I am pleased to inform you that your manuscript has been formally accepted for publication in PLOS Computational Biology. Your manuscript is now with our production department and you will be notified of the publication date in due course.

With kind regards,

Anita Estes
